# Deleting the C84L Gene from the Virulent African Swine Fever Virus SY18 Does Not Affect Its Replication in Porcine Primary Macrophages but Reduces Its Virulence in Swine

**DOI:** 10.3390/pathogens13020103

**Published:** 2024-01-24

**Authors:** Jinjin Yang, Rongnian Zhu, Yanyan Zhang, Xintao Zhou, Huixian Yue, Qixuan Li, Junnan Ke, Yu Wang, Faming Miao, Teng Chen, Fei Zhang, Shoufeng Zhang, Aidong Qian, Rongliang Hu

**Affiliations:** 1College of Veterinary Medicine, Jilin Agricultural University, Changchun 130118, China; 2Changchun Veterinary Research Institute, Chinese Academy of Agricultural Sciences, Changchun 130122, China; 3Key Laboratory of Prevention & Control for African Swine Fever and Other Major Pig Diseases, Ministry of Agriculture and Rural Affairs, Changchun 130000, China

**Keywords:** African swine fever virus (ASFV), C84L, deletion, virulence, replication

## Abstract

(1) Background: African swine fever (ASF) is a highly contagious disease that causes high pig mortality. Due to the absence of vaccines, prevention and control are relatively challenging. The pathogenic African swine fever virus (ASFV) has a complex structure and encodes over 160 proteins, many of which still need to be studied and verified for their functions. In this study, we identified one of the unknown functional genes, C84L. (2) Methods: A gene deficient strain was obtained through homologous recombination and several rounds of purification, and its replication characteristics and virulence were studied through in vitro and in vivo experiments, respectively. (3) Results: Deleting this gene from the wild-type virulent strain SY18 did not affect its replication in porcine primary macrophages but reduced its virulence in pigs. In animal experiments, we injected pigs with a 10^2^ TCID_50_, 10^5^ TCID_50_ deletion virus, and a 10^2^ TCID_50_ wild-type strain SY18 intramuscularly. The control group pigs reached the humane endpoint on the ninth day (0/5) and were euthanized. Two pigs in the 10^2^ TCID_50_(2/5) deletion virus group survived on the twenty-first day, and one in the 10^5^ TCID_50_(1/5) deletion virus group survived. On the twenty-first day, the surviving pigs were euthanized, which was the end of the experiment. The necropsies of the survival group and control groups’ necropsies showed that the surviving pigs’ liver, spleen, lungs, kidneys, and submaxillary lymph nodes did not show significant lesions associated with the ASFV. ASFV-specific antibodies were first detected on the seventh day after immunization; (4) Conclusions: This is the first study to complete the replication and virulence functional exploration of the C84L gene of SY18. In this study, C84L gene was preliminarily found not a necessary gene for replication, gene deletion strain SY18ΔC84L has similar growth characteristics to SY18 in porcine primary alveolar macrophages. The C84L gene affects the virulence of the SY18 strain.

## 1. Introduction

African swine fever (ASF) is an acute and highly contagious disease that causes hemorrhagic fever in pigs, with more than 100 years of history. The prevalence and spread of this disease have caused a heavy blow to the world’s pig industry and economics. The disease first appeared in Africa, was discovered in Georgia in 2007, and then spread to the Russian Federation, followed by Estonia, Latvia, Lithuania, Poland, and the Czech Republic and in other European countries, and was introduced to China in 2018 [1,2,3,4,5,6,7]. The current lack of safe and effective vaccines poses a great possibility for prevention and control. Improving the understanding of the unknown gene function of ASFV can further studies of the viral pathogenic mechanism, providing a reference for developing antiviral drugs and vaccines.

ASFV is a double-stranded DNA virus with a regular icosahedral structure, with a complex genome length of 173 to 193 kbp, varying among different isolates. Changes in multiple gene family genes ultimately lead to the emergence of different virus strains [8]. ASFV encodes over 160 proteins, of which known functional proteins play roles in viral structure, viral replication, and virus–host interaction. The icosahedron structure of ASFV consists of five layers: the core, the core–shell, the inner membrane, the nucleocapsid, and the outer membrane [9,10,11]. Each layer is composed of various proteins. The main protein in the inner core is p10, a DNA-binding protein [12]. The core–shell mainly comprises polymeric proteins pp220 and pp62, which can be cleaved by protein pS273R, a protease associated with ASFV infection [9,13,14,15]. The inner envelope originates from the endoplasmic reticulum and contains the main viral protein of p54, p17, and p12 [16,17,18]. The capsid structure of ASFV is primarily composed of the p72 protein, and another protein CD2V, a homolog of cell CD2 protein, mediates the blood adsorption function of ASFV. CD2V is located within the virus particles [19,20]. In addition, studies have reported that certain proteins of ASFV are associated with viral replication and the virus’s ability to evade the innate immunity of host. The protein encoded by the A104R gene influences viral replication and genome packaging, while the A151R protein is associated with viral replication and assembly [21,22]. Some proteins, such as the I329L protein, play a role in the life cycle of ASFV by inhibiting the production of IFN [23,24]. In addition, some proteins, such as A179L, inhibit the host cell’s apoptosis pathway to prevent the virus from escaping the host’s immune response [25].

Many studies have successfully verified the function of many genes by deleting one or more genes from the ASFV genome. The main research direction of these studies is verifying whether deleting this gene affects the replication of the virus in cells and its virulence in pigs. Several vaccine candidate strains have been identified, such as deleting the I226R gene or L7L-L11L from the virulent SY18 strain and the deletion of I177L from the Georgian strain [26,27,28], which provided some new target genes for studying the mechanism of viral infection. At the same time as conducting this study, other researchers in the same field have studied the structure and function of C84L. pC84L is a protein expressed in both the cytoplasm and nucleus in the middle and late stages. The expression of C84L protein can induce the activation of NF-κB and NLRP3 inflammasome mediated IL-1β production, leading to cell pyroptosis. In addition, the expression of C84L protein induces the transcription and secretion of pro-inflammatory cytokines such as IL-6, IL-12, IFN-β and TNF-α. Therefore, deleting the C84L gene in the wild strain of ASFV may inhibit the excessive activation of the inflammatory system and expression of inflammatory factors caused by the high virulence strain of ASFV, making it a vaccine candidate gene [29]. In this study, the C84L gene was deleted from virulent ASFV SY18 and the results showed that deleting the gene from the SY18 strain did not affect the replication of the virus in the primary macrophages of pigs. However, the virus’s virulence in pigs was reduced and this reduction displayed a certain dose dependence.

## 2. Materials and Methods

### 2.1. Cells and Viruses

The method for obtaining and culturing primary porcine macrophages was the same as previously described [26]. All PAM cells in this study were subjected to nucleic acid extraction for exogenous virus testing before use to ensure no other virus interference, just like other studies in this laboratory. The detected exogenous viruses include ASFV, classical swine fever virus (CSFV), porcine reproductive and respiratory syndrome virus (PRRSV), pseudorabies virus (PRV), porcine parvovirus (PPV), and porcine circovirus ½ (PCV1/2). At present, the preparation method has been quite mature in our laboratory. The culture medium for cultured cells was Roswell Park Memorial Institute (RPMI)-1640 (Gibco, Beijing, China), containing 10% fetal bovine serum (Gibco) and 1% penicillin-streptomycin solution Cells were cultured in a cell incubator at 37 °C and 5% CO_2_. The virus SY18 used in this study was isolated by the Military Veterinary Research Institute in 2018 and stored in a biosafety level 3 (BSL-3) laboratory [7]. The specific operation for the determination of the virus titer involved pre-seeding a single layer of porcine primary macrophages in a 96-well culture plate (Corning); then adding the virus solution to the culture plate with a 10-fold gradient dilution and culturing it in a cell incubator with 5% CO_2_ at 37 °C for 5 d; then dying it with the p30 monoclonal antibody prepared by our laboratory and observing the fluorescence under the microscope through a fluorescence microscope; then calculating the virus titer using the method of Reed–Munch [30].

### 2.2. Characteristic of the C84L Gene Expression

To determine the duration of the C84L gene during the virus infection cycle, we tested the mRNA expression levels of the C84L gene of SY18 at 1, 4, 8, 12, 16, 24 h post-infection, respectively. PAM cells were infected with ASFV SY18 at 3MOI. Samples were collected and calculated as described in the previous studies [31].

### 2.3. Construction of ASFV Recombinant Virus SY18∆C84L

Flanking DNA fragments mapping to the left (1.2 kbp) and right (1.2 kbp) of C84L were amplified by PCR using SY18 genomic DNA as a template. The forward primer for the left flank was 5′-CGCTTAAAACGGTATATCCAG-3′and the reverse primer was 5′-ACGTATATGAATTATTTTTTAGC-3′. For the right flank, the forward primer was 5′-ATAAATAGCAGCATATATTA-3′ and the reverse primer was 5′-CAATTAAGTACCCTAGTAAGTC-3′. The plasmid obtained by a homologous recombination of the left and right arm amplified fragments was ligated into the pMD18-T vector (modified in our laboratory), which contained p72 and mCherry expression cassettes and a shuttle plasmid pΔC84L-mCherry was obtained. PAMs were cultured in 6-well plates (Corning) infected with ASFV SY18 at a multiplicity of infection (MOI) of 1.0 and transfected with 2 µg of pΔC84L-mCherry using jetPEI^®^-macrophage transfection (Polyplus, Illkirch, France) two hours later. Selecting cells that emitted red fluorescence through 5–6 rounds of single-cell selection, followed by an additional 6–7 rounds of limited dilution, then the deletion mutant SY18ΔC84L was obtained.

### 2.4. Identification of Recombinant Virus and Next-Generation Sequencing

A pair of primers were designed on the C84L gene, using forward primer 84-F: 5′-CGTAAACCGTAATAAACATC-3′ and the reverse primer 84-R: 5′-TGGATCAGGAACAACTTTTC-3′. If the deletion virus did not exhibit a 148 bp electrophoretic band, the wild-type virus strain SY18 displayed a 148 bp electrophoretic band, which proved the purification of the deletion virus SY18ΔC84L. To verify the accuracy of the recombinant virus genome, the entire genome was sequenced using next-generation sequencing. A laboratory kit was used to extract the complete virus genome, which was subsequently sent the genome to a company for next-generation sequencing (Novogene Co., Ltd., Tianjin, China).

### 2.5. Growth Characteristic of SY18∆C84L In Vitro

To understand whether the deletion virus SY18∆C84L affects replication in primary macrophages of pigs compared to its wild-type strain SY18, we studied the growth characteristics of the two viruses and plotted growth curves. PAMs were cultured in a 24-well plate and the cells were infected with the deleted SY18∆C84L virus and the wild-type virus strains SY18 at an MOI of 0.01 (which allows for better detection of changes in growth trends). Subsequently, samples were collected at 2, 12, 24, 48, 72, 96, and 120 h after infection and subjected to three cycles of freezing and thawing. Finally, the viral titer of the samples was calculated through Reed–Munch, and the growth curves were plotted.

### 2.6. Animal Experiments

This animal experiment was conducted in the Biosafety Level 3 Laboratory of the Military Veterinary Research Institute and received approval from the Animal Welfare and Ethics Committee.

To identify whether deleting the C84L gene from ASFV affected its virulence in pigs, we prepared 15 pigs for the experiment, which were purchased from a local farm with high-quality breeding conditions. The experiment was divided into three groups named groups A, B, and C, each group consisting of 5 pigs, with different ear numbers used to distinguish individual animals. The numbers for group A are 49, 50, 51, 52, and 53. The numbers for group B are 55, 56, 57, 58, and 59. The numbers for group C are 01, 02, 03, 04, and 05.In group A, pigs were administered an intramuscular injection of 10^2^ TCID_50_ deletion virus SY18∆C84L. Group B was immunized with the deletion virus SY18∆C84L at a dose of 10^5^ TCID_50_ following a similar procedure. Group C was intramuscularly injected with the wild-type strain SY18 at a dose of 10^2^TCID_50_ as a control. After immunization, temperature and clinical symptoms were monitored daily. Blood, oral, and anal swabs from pigs were collected on day 0, 3, 7, 14, and 21 of immunization to detect the ASFV genome content in these samples. On the twenty-first day of the observation period, the surviving pigs were euthanized, and their tissues were collected for ASFV genome quantification and HE staining for pathological observation. These tissues included the heart, liver, spleen, lungs, kidneys, bone marrow, and various lymph nodes.

### 2.7. Quantitative PCR Analysis of Virus Genome Copies Numbers

The laboratory-established qPCR method was employed to detect experimental samples targeting the ASFV p72 gene, and the method and primer design referred to OIE standards. Forward primer 5′-CTGCTCATGGTATCAATCTTATCGA-3′, reverse primer 5′-GATACCACAAGATCAGCCGT-3′, and a TaqMan probe FAM-5′-CCACGGGAGGAATACCAACCCAGTG-3′-TAMRA were used. The specific experimental operating conditions were as described previously [28].In each testing experiment, the genomic content of ASFV in the sample was calculated using equations developed based on Cq values at different concentrations of the p72 gene standard plasmid.

### 2.8. Detection of Anti-ASFV Antibodies

We conducted an antibody test on the serum of immunized and infected pigs to detect whether the deletion virus had certain immunogenicity. Our laboratory developed a test kit targeting the p54 protein of ASFV. The operation step of this indirect enzyme-linked immunosorbent assay method was to first coat the purified p54 protein on a special ELISA 96-well plate for the experiment, place it in a refrigerator at 4 °C overnight, wash it with PBST three times, and then block it with 5% skimmed milk at 37 °C for 2 h. After washing it three times, we successively added the sample, as well as the negative and positive control sample, and incubated them at room temperature for 1 h. After washing 3 times, HRP labeled sheep anti-pig IgG was added and incubated at room temperature for 1 h. We continued to wash the solution 3 times, TMB color-developing solution was initiated and dyed at 37 °C for 8 min. Finally, it was terminated with 2M sulfuric acid and read under the OD_450_ condition of the enzyme marker. The ratio of the OD_450_ value of each sample to the OD_450_ value of the positive sample(S/P) exceeding 0.25 indicated the presence of ASFV antibodies. Detailed procedures can be found in our previous publication [27].

### 2.9. Biosafety Statement and Facility

All experiments on the ASF live virus were conducted at the biosafety level 3 of the Military Veterinary Research Institute and were approved by the Ministry of Agriculture and Rural Affairs and the China National Accreditation Service for Conformity Assessment. The experiments were approved by the Animal Welfare and Ethics Committee of the Changchun Veterinary Research Institute, Chinese Academy of Agriculture (Review ID: IACUC of CAS-12-2021-011, approved on 1 December 2021).

### 2.10. Statistical Analysis

Statistical analyses were conducted using unpaired, two-tailed Student *t*-test. A *p* value < 0.05 was considered statistically significant.

## 3. Results

### 3.1. C84L Gene Is Not Conserved across Different ASFV Isolates and Transcribed in the Early Stage

The ORF of C84L of ASFV SY18 strain is located on the reverse chain between 81,385–81,615 bp across the entire genome, encoding 76 amino acid residues (Figure 1A). To understand the conservation of the C84L gene, we downloaded the amino acid sequences of 10 pathogenic virus isolates in Europe, Africa, and the Caribbean on the NCBI website and performed multiple amino acid sequence alignments using the MAFFT online website and Jal view software 2.11.1.5 (http://www.jalview.org/). The results presented in Figure 1B demonstrate that most of the virus isolates encode C84L protein with 72 amino acids. These findings showed that the C84L gene exhibits low conservation among different isolates.

Using the early expression protein p30 and late expression protein p72 of ASFV as controls, p30 had the highest expression level at the eighth hour of the virus infection cycle, p72 at the sixteenth hour, and C84L had the highest expression level after eight hours post infection, similarly to p30 (Figure 1C). The results indicate that C84L is transcribed in the early stages of the virus infection cycle.

### 3.2. Generation of SY18ΔC84L

PAMs were infected for 2 h and transfected 2 μg of plasmid p∆C84L mCherry (Figure 2A). Six hours later, the cell culture medium was replaced with a fresh medium, and cells with red fluorescence, known as the deletion virus SY18ΔC84L, could be seen under a fluorescence microscope within sixteen hours. After that, 3–5 rounds of single cells with red fluorescence were selected and subjected to several rounds of limited dilution. The deletion strain SY18ΔC84L was obtained by PCR amplifying the C84L gene for verification (Figure 2B). PCR identification and next-generation sequencing of the obtained deletion virus showed that the C84L gene was replaced by an mCherry expression cassette with p72 gene promoter.

### 3.3. Replication of Recombinant Deletion Virus on Primary Porcine Macrophages

To verify whether the deletion of the C84L gene affected the replication of SY18 in vitro, porcine primary macrophages were infected with the deletion virus SY18ΔC84L and the wild-type virus SY18 at a multiplicity of infection (MOI) of 0.01, respectively. Cell cultures were collected at different infection times (2, 12, 24, 48, 72, 96, 120 h) and freeze–thawed thrice. The supernatants were centrifuged, and we determined the viral titer using the Reed–Muench method [29]. The results indicated that there were no changes in the growth characteristics of the deletion virus SY18ΔC84L and its wild-type strain SY18 on the porcine primary macrophages (Figure 2C). Virus replication reached a plateau or declined after 96 to 120 h post infection, suggesting that the deletion of the C84L gene in ASFV did not affect the replication of the virus.

### 3.4. Virulence and Pathological Characteristics of the SY18ΔC84L in Pigs

Three groups were designed for this experiment to evaluate the virulence of the deletion mutant SY18ΔC84L in pigs. Five pigs in each group were intramuscularly injected with either 10^2^ TCID_50_ (group A) or 10^5^ TCID_50_ (group B) per pig. The SY18 strain was also intramuscularly injected with 10^2^ TCID_50_ (group C) per pig as a control. As expected, on the fifth day after inoculation, the body temperature in group C began to rise, while the body temperature in group B, inoculated with 10^5^ TCID_50_/pig deletion mutant SY18ΔC84L, also began to rise. Pigs vaccinated with 10^2^ TCID_50_ ASFV SY18 strain developed the disease within ten days and were euthanized to the endpoint on the ninth day (Figure 3A,B and Table 1). From the sixth day, except for the pig No. 53 in group A, whose body temperature was normal, all other pigs showed fever symptoms. Pig No. 50 in group A, which survived for 21 days, has a low fever state (body temperature < 41 °C). On the twenty-first day of the observation period, the body temperature of pig No.56 in group B returned to normal levels (Figure 3B). In the experiment, all non-surviving pigs exhibited clinical symptoms such as fever, anorexia, and depression before being euthanized. Compared to control group C, pigs in groups A and B had a longer course of disease and more obvious onset symptoms (Table 1).

Blood, oral swabs, and anal swabs were collected from experimental pigs on day 0, 3, 7, 14, and 21 after vaccination and quantified the genome of ASFV using quantitative PCR. Control group C detected the ASFV genome in the blood on the third day, and the genome content increased on the seventh day. On day 3, only pig No. 52 had a higher ASFV genome content in the blood of group A, while other pigs remained at very low levels. On day 7, the ASFV genome content increased, but the ASFV genome content in pigs No. 51 and 53 remained relatively low. The pigs in group B detected a higher level of ASFV genome content from the third day, except for the ASFV genome content in pig 56, which remained lower during the observation period (Figure 3C). Compared with group C, lower levels of ASFV genome content were detected on oral and anal swabs from group A and group B pigs on the 7th day (Figure 3D,E). Therefore, the risk of virus shedding was relatively low.

In order to detect the distribution of the ASFV in the tissues of pigs after infection, qPCR was used to detect the viral genome content. The samples were collected from the heart, liver, spleen, lungs, kidneys, and submandibular lymphoid tissues of three groups of pigs previously described. The time of sample collection is consistent with the time described in the survival curve (Figure 3A), and the tissues of pigs that lived to 21 days were collected after euthanasia. The results showed that high viral genome content was detected in all tissues, whether they were inoculated with SY18 or SY18ΔC84L (Figure 4).

To study the pathological changes of surviving pigs immunized with different doses of SY18ΔC84L clearly, samples of pigs that survived after 21 days of immunization were collected and used to prepare pathological sections, with tissue pathological sections of pigs that died on the ninth day after challenging with SY18 as control. Similar to previous reports, the pigs challenged with SY18 showed varying degrees of hyperemia and multifocal diffuse hemorrhage in all tissues (Figure 5) [26]. The multifocal infiltration of several inflammatory cells is mainly seen in lung tissue, and spleen and lymph nodes show a significant decrease in lymphocytes and nuclear fragmentation. Structural changes and extensive cellular atrophy were appeared in kidney tissue (Figure 5). The liver and kidney of pigs immunized with low-dose SY18ΔC84L showed slight pathological changes, while the pathological changes in other tissues were not significant. More interestingly, all tissues of the surviving pigs inoculated with high-dose SY18ΔC84L showed no significant pathological changes (Figure 5). The above results indicate that the deletion of C84L gene from SY18 can attenuate its virulence in pigs.

### 3.5. Antibody Reaction of the SY18ΔC84L

To evaluate the immune response of the deletion strain to pigs, we collected animal serum at 0, 3, 7, 14, and 21 dpi and used indirect ELISA to detect the antibody level against ASFVp54 protein in the serum. The results showed that on the seventh day, only pigs labeled No. 52 in group A immunized with 10^2^ TCID_50_ deletion virus SY18ΔC84L produced antibodies. On the fourteenth day, the antibody levels of surviving pigs increased; on the twenty-first day, a higher antibody level was detected in the surviving pigs’ bodies. On the seventh day, all five pigs in group B were immunized with 10^5^ TCID_50_ deletion virus SY18ΔC84L produced antibodies. On the fourteenth day, the antibody levels in the surviving pigs increased, and on the twenty-first day, the antibody levels in the surviving pigs were still detectable, but there was no significant change compared to the antibody levels on the fourteenth day. However, the pigs in group C vaccinated with the wild-type strain ASFV-SY18 did not produce antibodies on the seventh day (Figure 6).

## 4. Discussion

The African swine fever has caused serious losses to the world’s pig industry. At present, the use of genetic engineering technology to develop attenuated live vaccines has great development prospects. However, many unknown gene functions have yet to be elucidated due to the complexity of the structure and function of the protein encoded by the ASFV genome. Therefore, clarifying the function of unknown genes in ASFV can provide some reference for understanding the virus’s infection and pathogenesis and developing safe and effective vaccine-candidate strains.

With researchers’ continuous study on ASFV in recent years, the various functions of the ASFV gene have gradually been elucidated. Deleting the C962R or E165R genes from the wild-type strain did not affect the replication of the virus in primary macrophages, nor did it affect the virulence of the virus in pigs [32,33]. Deleting similar genes can serve as a continuous fluorescent marker for viruses to investigate further the molecular mechanisms of the pathogenicity of the strain [34]. In addition, there were also some genes related to the virulence of ASFV. Deletion from the wild-type strain could reduce virulence in pigs [35,36]. Based on numerous experimental results, many virulence genes related to some vaccine candidate strains have also been reported [26,27,28,37,38,39]. Although natural attenuated strains also have a certain role in epidemic prevention and control, considering safety issues, the most likely vaccine to be launched is the attenuated live vaccine lacking genetic engineering technology.

Our current research results indicate that the virulence of the virus with the deletion of C84L gene in pigs was partly reduced and had a certain dose dependence. After 21 days of observation, the body temperature of surviving pigs immunized with a low dose deleted mutant virus decreased but remained in a persistent low fever state. However, the body temperature of surviving pigs immunized with high doses decreased and returned to normal. This phenomenon may be related to the inflammatory reaction occurring in the pig body and is worth further research. The viremia in the blood of pigs with immune deletion virus can be detected from the third day, and its content is lower than that of pigs vaccinated with the SY18 group, while the viremia content of surviving pigs immunized with high-dose deficient virus has always been at a very low level. The viral load in the oral and anal swabs of immune deletion virus pigs has always been at a very low level compared to pigs inoculated with SY18, and some pigs even have negative viral load in their oral and anal swabs. During the autopsy, the ASFV genome content in various tissues and organs of experimental pigs was detected. It was found that the virus genome content in various tissues and organs of pigs inoculated with the SY18 group was higher than that of pigs immunized with the same dose of the SY18ΔC84L group. Results indicate that deleting the C84L gene from ASFV SY18 weakens the viral replication in pigs, decreasing viral virulence in pigs.

In this study, the antibodies of pigs infected with an immunodeficiency virus were detected on the seventh day. Some pigs infected with the immunodeficiency virus produced antibodies but eventually developed the disease and did not survive. This research result is similar to previous research reports. There is no direct correlation between the antibodies produced by animals infected with attenuated viruses and the immunity of the virus and the host. In other words, the specific mechanism has yet to be elucidated [40]. ASFV infection can also induce host cellular immune responses according to reports. However, the cellular immunity triggered by the attenuated strain is more obvious, and pigs infected with the virulent strain die before producing significant cellular immunity. Previous studies have shown that the proliferation of CD8α-T cells was only detected on the tenth day after ASFV infection in pigs [41]. In this study, pigs immunized with SY18 C84L began to die on the eighth day. However, the expression of C84L protein can induce an increase in TNF-a levels in PAM cells, indicating that C84L protein is also associated with cellular immunity according to recent C84L functional studies [29].

The prevention and control of ASF remains a global challenge, and the ASFV-G- ∆I177L deletion strain is currently the closest to becoming a commercialized vaccine. The ASFV-G-∆I177L deficient strain can protect pigs from attacks from parental virulent strains. Longer lasting experiments have shown that pigs vaccinated with ASFV-G-∆I177L have low levels of viremia, no virus shedding, and can produce higher levels of ASFV specific antibodies [28,42]. In this study, the survival rate of pigs vaccinated with SY18-C84L did not reach 100%. However, the surviving pigs produced specific antibodies against ASFV during the 21-day observation period, indicating that the absence of C84L, a virulence related gene, can only partially attenuate ASFV SY18. However, further deletion of other virulence related genes can be carried out on the basis of this study to further attenuate ASFV and protect pigs from challenges from virulent strains. Further research related to this is still ongoing.

We deleted the gene C84L from the highly virulent strain ASFV SY18 and constructed the deletion virus SY18ΔC84L. The replication ability of the deletion virus strain in porcine primary macrophages was similar to that of its wild-type strain SY18, and it could attenuate the virus’s virulence in pigs with a certain dose dependence. We plan to conduct virus challenge experiments on infected and surviving pigs and further investigate the virus infection mechanism in the next step.

## 5. Conclusions

This is the first study to complete the replication and virulence functional exploration of the C84L gene of SY18. In this study, C84L gene was preliminarily found not a necessary gene for replication, gene deletion strain SY18ΔC84L has similar growth characteristics to SY18 in porcine primary alveolar macrophages. The C84L gene affect the virulence of the SY18 strain. The gene deletion strain SY18ΔC84L had a certain pathogenicity to pigs, but after 21 days of observation, some pigs can still survive, and there was a certain dose dependence. Whether surviving pigs can resist the attack of the SY18 strain is the next step of research.

## Figures and Tables

**Figure 1 pathogens-13-00103-f001:**
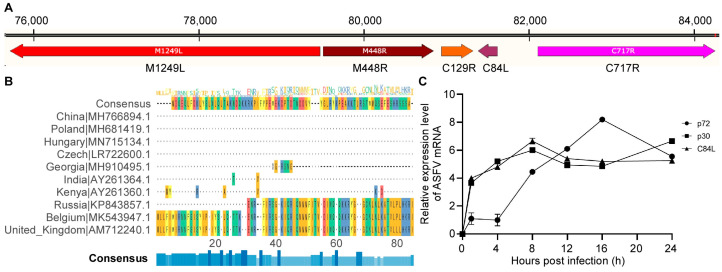
C84L gene is not conserved across different ASFV isolates and transcribed as an early viral gene. (**A**) Location of C84L gene. (**B**) Comparison of C84L protein between ASF isolates from different regions. Missing genes are represented by ‘-’. (**C**) Expression levels of mRNA of C84L, CP204L, and B646L. The values of the Y axis were expressed by the base-10 logarithm (log10) of the relative expression level.

**Figure 2 pathogens-13-00103-f002:**
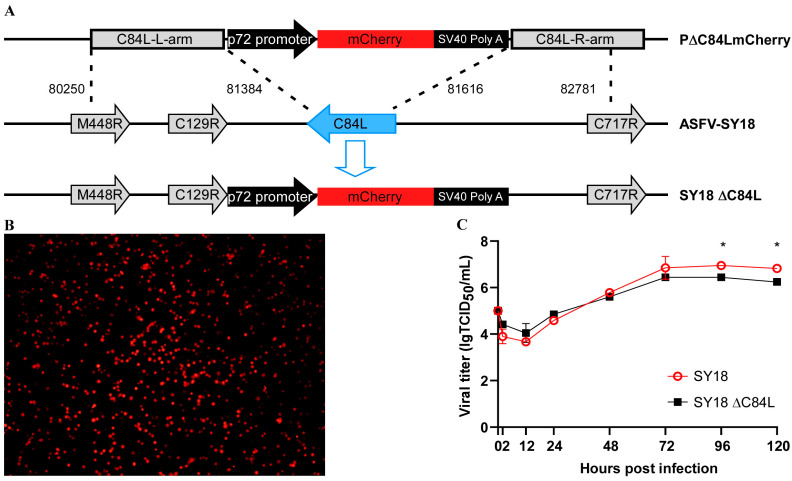
Construction of SY18ΔC84L and its infection in PAM cells. (**A**) The C84L gene fragment of ASFV was replaced with an mCherry expression box, as shown in the construction diagram. (**B**) PAM cells were infected with the deletion virus SY18ΔC84L showing mCherry expression. (**C**) In vitro growth characteristics of SY18ΔC84L and SY18. Primary macrophages were infected with these two viruses (MOI = 0.01) and titrate the virus titers at different time periods. Data represent means from three independent experiments. * *p* < 0.05.

**Figure 3 pathogens-13-00103-f003:**
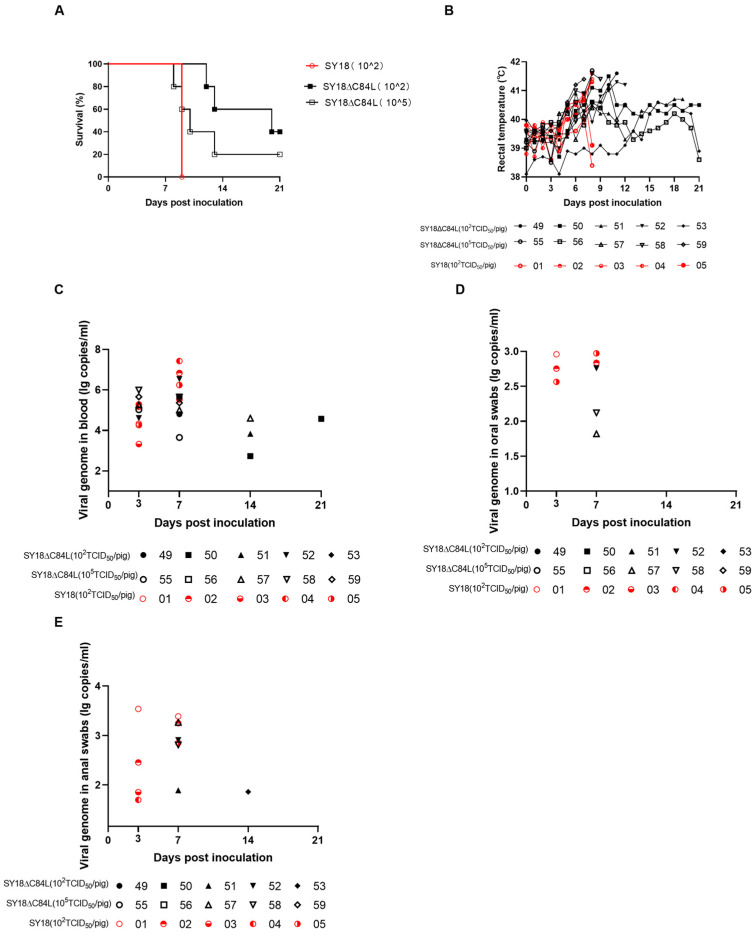
Virulence evaluation of SY18ΔC84L and SY18 in pigs. (**A**) Survival and body temperature (**B**) in pigs challenged with SY18ΔC84L and wild-type virus SY18. The genomic content of ASFV in the blood (**C**), oral swabs (**D**), and anal swabs (**E**) was detected in pigs after inoculation with SY18ΔC84L and SY18 at different doses.

**Figure 4 pathogens-13-00103-f004:**
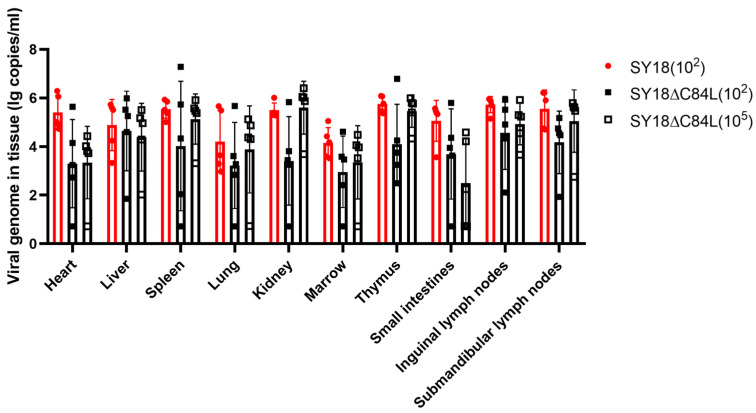
ASFV genome content of different pig tissues. ASFV genome content of different pig tissues were detected.

**Figure 5 pathogens-13-00103-f005:**
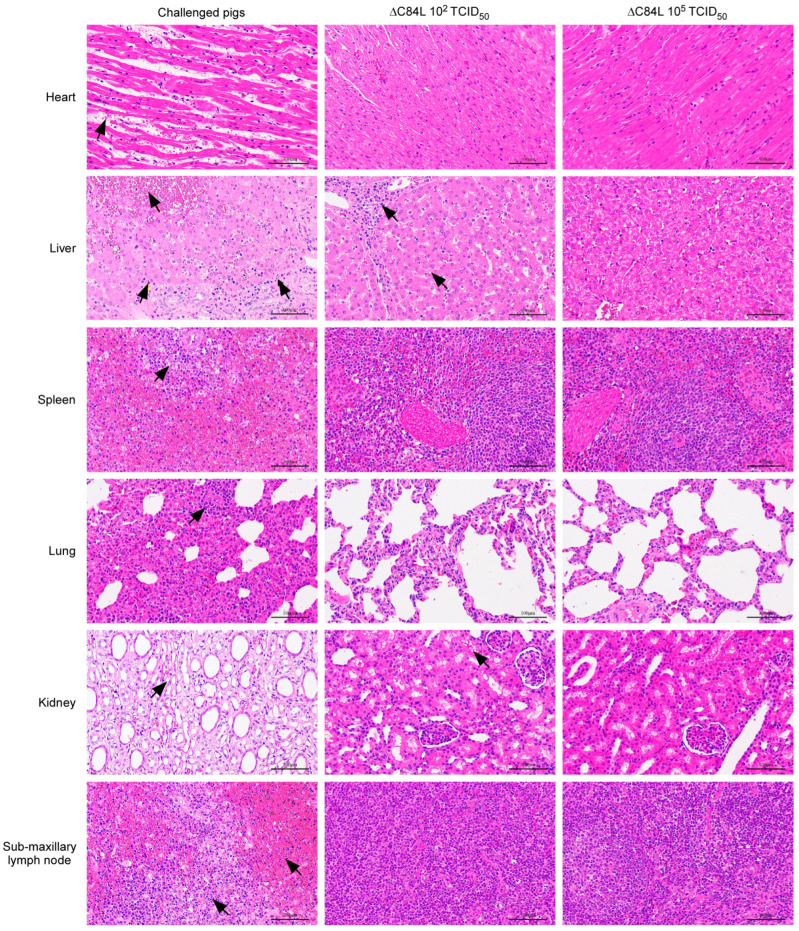
Characterization of histopathological lesions of different organs from indicated pigs challenged with ASFV SY18. The piglets were challenged with SY18 and different dose of SY18ΔC84L. The different tissue samples were collected as indicated to analyze the comparative histopathological lesions. The images show comparative histopathological lesions in different tissue samples. Arrows indicate pathological changes in relevant tissues.

**Figure 6 pathogens-13-00103-f006:**
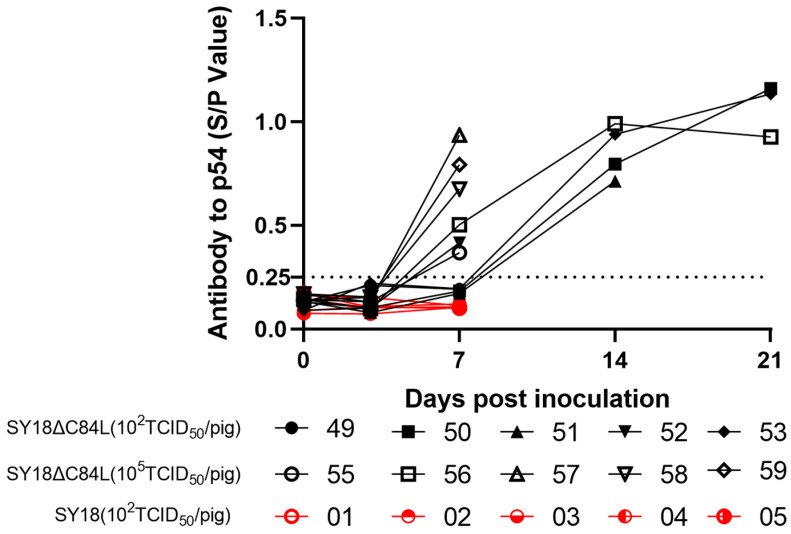
Antibody detection in pig serum. The African Swine Fever indirect ELISA method detected specific antibodies in pig serum. The dashed line represented a threshold of 0.25.

**Table 1 pathogens-13-00103-t001:** Swine survival and fever response following infection with different doses of SY18ΔC84L and parental ASFV SY18.

Virus and Dose (TCID50)	No. of Survivors/Total	Mean Time to Death (Days ± SD)	Fever		
			No. of Days to Onset (Days ± SD)	Duration No. of Days to Onset (Days ± SD)	Maximum Daily Temp (°C ± SD)
SY18-10^2^	0/5	9 (0)	5.75 (0.83)	3 (1.22)	41.3 (0.25)
SY18ΔC84L-10^2^	2/5	15 (3.56)	11.4 (3.77)	3.6 (0.49)	41.14 (0.41)
SY18ΔC84L-10^5^	1/5	10 (1.87)	6.8 (1.94)	3.2 (0.75)	41.28 (0.40)

## Data Availability

The data are contained within the article.

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
