# Peer review of "Deleting the C84L Gene from the Virulent African Swine Fever Virus SY18 Does Not Affect Its Replication in Porcine Primary Macrophages but Reduces Its Virulence in Swine"

_pathogens, 2024, doi:10.3390/pathogens13020103_

Round 1

Reviewer 1 Report

Comments and Suggestions for Authors

The study by Yang et al. investigated the impact of C84L on virulence and viral pathogenesis in pigs. This is the first-of-this-kind report of viral attenuation by C84L gene deletion and findings in this study could potentially lay the foundation for developing ASF live vaccine. The manuscript is well-written with minor errors and a good fit for publication on this journal. Major and minor comments pertaining to the study are as below.

Major comments:

- The main argument is the novelty of C84L gene founding and deletion. In this case, more details should be given. This can be improved by structure modeling of the protein, describing if this is a structure protein or not, etc. Investigating its role in immune antagonism is essential for creating gene-deletion strains, but these provides some basic infor. of the protein.

- Since future directions include viral challenge studies to assess vaccine efficacy, it should be extensively discussed from the several perspectives. First, what are the parameters (viral shedding blood/tissue, symptoms, immune response, etc) for a efficacious live ASF vaccine? Check the I177L paper. Second, Although T cell response is not assessed in this study, it should be discussed here along with its role in protection.

Minor comments:

- Check "CO2" across the maintext and "2" should be subscript.

- Line 193 please delete one parentheses.

- Figure 2 has two duplicate subfigures.

Author Response

Response to Reviewer 1 Comments

1. Summary

Thank you for your efforts in improving the quality of our manuscript. We have carefully read the suggestions of the academic editor and two reviewers, and we have responded to our comments one by one as follows.

2. Point-by-point response to Comments and Suggestions for Authors

Comments 1: The main argument is the novelty of C84L gene founding and deletion. In this case, more details should be given. This can be improved by structure modeling of the protein, describing if this is a structure protein or not, etc. Investigating its role in immune antagonism is essential for creating gene-deletion strains, but these provides some basic infor. of the protein.

Response 1: Thanks for your advice. When we studied the function of C84L gene by constructing gene deletion strains, colleagues from other research institutions also studied the function of C84L at the same time, although our research on the function of C84L protein is still ongoing. In view of this, it is more necessary for us to publish our existing research. However, we will cite newly published paper in the “Introduction” to introduce the function of C84L protein to compensate for this deficiency. (Line 72-83)

It is a Chinese research article, and we have attached the link in reference format and Chinese format here. The references in English format have also been inserted in the correct position in our manuscript.

Reference: https://doi.org/10.16656/j.issn.1673-4696.2024.0008.

Comments 2: Since future directions include viral challenge studies to assess vaccine efficacy, it should be extensively discussed from the several perspectives. First, what are the parameters (viral shedding blood/tissue, symptoms, immune response, etc) for a efficacious live ASF vaccine? Check the I177L paper. Second, Although T cell response is not assessed in this study, it should be discussed here along with its role in protection.

Response 2: Thanks for your advice. We carefully read the research on I177L and evaluated the immune efficacy of C84L deficient strains based on the immune efficacy of I177L deficient strain and compared them in the discussion. In addition, we have also discussed the cellular immunity induced by ASFV and the cellular immunity associated with C84L according to your suggestion. (Line 367-387)

Minor comments:

Comments 3 Check "CO2" across the maintext and "2" should be subscript.

Response 3: Corrected.

Comments 4 Line 193 please delete one parentheses.

Response 4: Corrected.

Comments 5 Figure 2 has two duplicate subfigures.

Response 5:  We have checked the section about Figure 2 and did not find any duplicate subgraphs. Please provide a more detailed explanation.

Reviewer 2 Report

Comments and Suggestions for Authors " The paper entitled as “Deleting the C84L Gene from the Virulent African Swine Fever 2 Virus SY18 Does Not affect its Replication in Porcine Primary 3 Macrophages but Reduces its Virulence in Swine” deals with an important infectious disease of swine that it has serious negative impact in swine production in many European, Asian, and African countries while it threads the worldwide pig production. Up to date there is a gap to the pathogenic mechanisms of the African Swine Fever virus (ASFv) while no treatment nor vaccine are available, thus the main control of the disease is based mainly on the implementation of strict biosecurity measures. The current paper provides valuable data in terms of pathogenic mechanisms of ASFv, and even more importantly in the scientific effort referring to the development of an effective vaccine. The authors were conducted research that focused on deleting specific genes from wild-type strain of ASFv, supplementing with important and new information the relevant conclusions of other researchers’ efforts in this specific topic. The different sections of the paper describe in an adequate way the research in terms of methodology used, the presentation of results through the tables and graphs is clear, the discussion is comprehensive, while the references used are the appropriate."

Line 40: Please add “and in other European countries”

Lines 57 and 357: Please leave blank after the dot

Author Response

1. Summary

Thank you for your efforts in improving the quality of our manuscript. We have carefully read the suggestions of the academic editor and two reviewers, and we have responded to our comments one by one as follows.

2. Point-by-point response to Comments and Suggestions for Authors

Comments 1: Line 40: Please add “and in other European countries”

Response 1: Corrected.

Comments 2: Lines 57 and 357: Please leave blank after the dot

Response 2: Corrected.